# Health and Medical Issues in the Area Affected by Fukushima Daiichi Nuclear Power Plant Accident

**DOI:** 10.3390/ijerph19010144

**Published:** 2021-12-23

**Authors:** Akemi Miyagawa, Koichi Tanigawa

**Affiliations:** Futaba Medical Center, Tomioka 979-1151, Japan; miyagawa@futaba-med.jp

**Keywords:** nuclear accident, evacuation, recovery, medical system, medical needs

## Abstract

Futaba County was the area most affected by the 2011 Fukushima Daiichi nuclear power plant accident. To understand issues around the re-development of the medical system, we investigated the post-accident changes in medical needs and the system’s transition. We analyzed reports from Fukushima Prefecture and local municipalities, ambulance transport data from the Futaba Fire Department, and patient data from Futaba Medical Center (FMC). After the accident, all medical institutions were closed, and the number of ambulance use dropped sharply. With the lifting of evacuation orders beginning in 2014, the amount of ambulance use increased at an annual rate of about 10%. Early on, the proportion of trauma caused by occupational and traffic accidents increased rapidly to more than 30%. As residents returned, the proportion related to endogenous diseases (most commonly respiratory) increased. Soon after the FMC opened in 2018, the majority of the patients were in their 60s, and by 2019 the proportion of patients in their 80s markedly increased. The return of the residents as well as ongoing decontamination and reconstruction projects were related to changes in the demographics of patients and the types of injuries and illnesses observed.

## 1. Introduction

Futaba County consists of eight towns and villages (the towns of Namie, Futaba, Okuma, Tomioka, Naraha, and Hirono, and the villages of Kawauchi and Katsurao) and is located on the eastern coast of Fukushima Prefecture. This was the area most affected by the accident at the Fukushima Daiichi nuclear power plant (NPP). The population of Futaba County was about 74,000, in 2011. After the accident, towns and villages within a radius of 20 km from the Fukushima Daiichi NPP and in the northwest were designated as an evacuation area, and all residents living in that area were evacuated [1]. Medical institutions and nursing care facilities in the area were also required to evacuate. Even at nearby medical institutions and facilities that escaped evacuation, medical services had to be limited because of a reduction in staff, due to concerns about radiation, and the reduced inflow of logistics and a shortage of necessary supplies [2].

As the restoration work at the Fukushima Daiichi NPP and the environmental decontamination progressed, the evacuation order gradually began to be lifted in 2014, and most recently in May 2019 was lifted in Okuma Town, where part of Fukushima Daiichi NPP is located [3]. In Futaba Town, which also contains part of the Fukushima Daiichi NPP, the evacuation order was lifted in a limited area, but residents are not yet allowed to return (Figure 1). After the evacuation order was lifted, the volume of traffic on the main road increased. People gradually returned, and as of April 2021, Futaba County had a resident population of 14,700 [4] (Figure 2). In addition, tens of thousands of workers were engaged in projects, such as decommissioning and decontamination work, the construction of interim storage facilities, reconstruction projects, and re-development of living infrastructure. It is predicted that medical needs will expand and diversify as the number of people moving in from outside the region increases.

Butler investigated 221 NPPs, and reported that about two-thirds of them had a population within a 30 km radius that is greater than that of the Fukushima Daiichi NPP, and there are 21 NPPs with a local population of over 1 million [5]. In the event of a major nuclear accident, it is expected that many residents will be forced to evacuate, but there will be areas to which they will return and resume their lives. The recovery process for every accident is different because it is determined by the type and magnitude, the capacity of governments to manage it, as well as the local context and circumstances. The ultimate goal of health systems recovery is to design a system that is able to respond to the demands and health needs of the population; perform its functions effectively, efficiently, and sustainably; increase the resilience of health systems; and mitigate the risk of future health emergencies [6]. Until the Fukushima accident, there had been no major NPP accident in which an evacuation order was issued and where the area was later re-developed to allow residents to return. This study analyzes documents and data, to investigate the changes in medical needs after the accident, and reports issues in the recovery of the medical system. It can therefore help to formulate recovery plans for medical systems after any major nuclear accidents that occur in the future.

## 2. Methods

### 2.1. Design and Data Acquisition

To understand the changes in the medical system in Futaba County, we used the reports of the study panels on medical care and welfare in the evacuation area sponsored by Fukushima Prefecture [7]. The study panel has been held regularly since 2015, to discuss how to develop the medical system in Futaba County. The latest data were obtained from the 13th study panel, in February 2021. For data on changes in the resident population, we used the information on the returning residents reported by eight towns and villages in Futaba County, in April 2021 [4]. To assess emergency medical needs, we analyzed the annual data on ambulance use from 2010 to 2020, provided by the Futaba Fire Department. This data set was in line with the national format for the ambulance transport data. For medical data after 2018, we analyzed patient data from the Futaba Medical Center (FMC), which have been reviewed monthly since the establishment of the center. The data were verified as accurate in October 2021.

### 2.2. Statistical Analysis

For the ambulance data, we compared age groups, reasons for ambulance use, transport time to hospital before and after the accident, the cancellation of evacuation orders, and the establishment of FMC. In the FMC patient data, yearly comparisons were made for outpatients and inpatients by diagnosis based on the International Classification of Diseases (ICD). The chi-square (χ2) was used, and data were analyzed using SPSS version 28 (IBM, Armonk, NY, USA).

## 3. Results

### 3.1. Impact of the Fukushima Daiichi NPP Accident on the Futaba County Medical System

Before the Fukushima accident, Futaba County was dealing with the challenges of aging residents, depopulation, and a shortage of medical resources. The two core hospitals were therefore integrated to consolidate medical resources, and were preparing to provide services in this new from when the Great East Japan Earthquake and Fukushima Daiichi NPP accident occurred [8].

There were 48 medical clinics and 26 dental clinics in operation in Futaba County before the accident, but all medical institutions in the evacuation area stopped providing medical services after the accident. Only three medical clinics continued to provide medical care in the non-evacuated area of Futaba County [7]. As of July 2016, only seven medical clinics and three dental clinics had resumed medical services. Five of the six hospitals that were in operation before the accident were located in the evacuation order area; only one hospital with long-term/psychiatric beds escaped the evacuation order and continued to provide care. As a result, hospitals with emergency capability became completely absent in Futaba County, and the majority of emergency cases that occurred in the area were transported to medical institutions outside the region.

Figure 3 shows the emergency medical services (EMS) data from before and after the accident. In 2010, before the accident, the annual number of emergency use was 2454, but, in 2011, when the accident occurred, it dropped sharply to 794 (185 cases after 11 March of that year). The number of ambulance transports has gradually increased since 2013. The evacuation order for Kawauchi Village was lifted in 2014, for Naraha Town in 2015, and for Tomioka Town and Namie Town in 2017. With the lifting of the evacuation orders from these municipalities occurred, the number of ambulance use increased at an annual rate of about 10%, until 2019. Looking at the proportion of patients by age group, the number of young (age < 18 years old) and older people (age ≥ 65) decreased after the accident compared with before the accident, but, after 2017, when the return of the residents of Namie and Tomioka Town began, the proportion of older people increased (Figure 4). Regarding the reasons for ambulance use, the proportion of trauma caused by occupational accidents increased four-fold, in 2011, compared with before the accident. Furthermore, the proportion of traffic accidents increased from 2014, when the evacuation order began to be lifted (Figure 5). Since 2018, the proportion of traffic accidents has been decreasing, whereas the proportion of endogenous diseases has increased.

In Futaba and Okuma Town, in which the Fukushima Daiichi NPP is located and the lifting of the evacuation order has been delayed, the number of ambulance use after the accident was small (Figure 6). However, in Naraha, Tomioka, and Namie Town, in which evacuation orders were lifted earlier, the number of ambulance use increased as the returning population increased. In Hirono Town, in which sheltering in place was ordered, the ambulance use decreased but had recovered to pre-accident levels by 2015. In Kawauchi Village, in which the evacuation order was only issued for part of the village, the number of ambulance use temporarily decreased immediately after the accident, but no major changes were observed thereafter.

Figure 7 shows the changes in the average time required from emergency call (119 in Japan) to arrival at hospital (defined as accommodation of a patient by a hospital). Before the accident, the biggest group arrived at hospital within 30 to 60 min, but, after the accident, the proportion taking 60 to 120 min increased to about 70%, twice that before the accident. This reflects the poor access to hospitals, even in cases of acute trauma or emergency illnesses that require urgent care. In 2017, it took an average of 75.7 min from the 119 call to hospital arrival, which was nearly twice as long as the national average (39.3 min) [9]. More than 60% of ambulance transports took 60 to 120 min from the 119 call to hospital arrival (national average: 9.0%), and 7% took 120 min or more (national average: 0.4%).

The in-service transportation rate (the ratio of the number of patients transported to medical institutions in Futaba County out of the total number of ambulance use in the Futaba County) was 29% the year before FMC was opened; after the opening of FMC, it increased to 61% (the in-service transportation rate before the accident was 63%). In addition, the ratio of the number of patients transported to FMC out of the number of patients transported to medical institutions in Futaba County was 88.3% in 2018, 91.8% in 2019, and 91.6% in 2020. The proportion of ambulance transports that required 60 min or more to arrive at the hospital was 64.1% in 2017, but, after the opening of FMC, decreased to less than 50% (49.9%, 44% and 45.9% in 2018, 2019, and 2020, respectively).

### 3.2. Improvement in the Medical System in Futaba County: Establishment of FMC

Many evacuees cited the development of medical institutions as a requirement for their return. To meet these demands, the Japanese government and Fukushima Prefecture have supported the reopening and opening of clinics. In the towns of Naraha, Tomioka, and Namie, clinics were opened as the evacuation order was lifted. However, private medical institutions with beds were unable to re-open or resume providing medical services because of difficulties in securing human resources, underdeveloped infrastructure, uncertainties in medical demand, and problems with profitability in Futaba County. In April 2018, FMC, which is operated by Fukushima Prefecture, was opened to meet the increasing need for emergency medical care in Futaba County. FMC is a small hospital with 30 acute beds, established to provide initial emergency medical care, including responses to nuclear and radiological emergencies at the NPP. It also runs a medical helicopter to eliminate geographical disadvantages, such as mountainous terrain. Although its main role is to provide emergency medical care, it also has additional functions, such as the provision of home care for older patients and support of health promotion for residents.

Figure 8 shows the changes in emergency outpatient visits over time. Since its establishment, the number of patients treated at FMC has increased. In the fiscal year (FY) 2019 (in Japan, FY starts in April and ends in March), it increased by 50% over the first year of operation. From November 2020 to February 2021, the number of patients decreased, but, from March 2021, the number of patients began to increase again.

Figure 9 shows the changes in the number of patients classified by their health insurance card address. In FY2018, patients with an address outside Futaba County accounted for more than 30% of the total, but, after that, the number of patients from all areas of Futaba County increased, except for Futaba Town and Katsurao Village. The rate of increase was 1.5 times for Naraha and Tomioka Town. In 2019, the evacuation order for a limited area in Okuma Town was lifted and patient visits from there increased. The number of patients with an address in Fukushima Prefecture, other than Futaba County, is also increasing. However, the number with an address outside the prefecture peaked in FY2019 and has decreased since then.

The largest group of patients in the first year of operation was those in their 60s. The number of patients in all age groups increased in FY2019, but the rate of increase in patients in their 80s was remarkable, more than doubling (Figure 10). In FY2020, the number of people aged 20 and under and those in their 40s decreased, and the increase in other age groups also slowed; however, the number of patients aged 90 and over increased twofold.

Looking at the ICD of outpatients who were treated at FMC, more than 30% were related to trauma (S, T: injury, poisoning, and certain other consequences of external causes; V, W: external causes of morbidity and mortality); for example, from occupational accidents and traffic injuries. However, J (diseases of the respiratory system) was the most common endogenous disease (Figure 11). There was a declining trend in J until FY2020. The patient volume was small, but the number of cases classified as E (endocrine, nutritional, and metabolic diseases) and F (mental and behavioral disorders) increased in FY2020. The former was most commonly diabetes mellitus, and the latter included 23 cases in FY2018 and 24 cases in FY2019, but increased to 54 cases in FY2020. These were most commonly F4 (neurotic, stress-related, and somatoform disorders) (26 cases), F2 (schizophrenia, schizotypal, and delusional disorders), F3 (mood (affective) disorders) (5 cases in total), and F1 (mental and behavioral disorders due to psychoactive substance use) (7 cases).

The number of inpatients increased nearly twofold in the 3 years after the opening of FMC (Figure 12). Classified by age, more than half of the patients were aged 70 and over (62%, 52%, and 61% in FY2018, FY2019, and FY2020). In FY2020, the increase in patients in their 80s was remarkable. The numbers of inpatients classified S, T, and J decreased slightly, while hospitalization due to K (diseases of the digestive system) increased (Figure 13). Enteritis and biliary tract diseases were commonly observed gastrointestinal diseases, and the acute exacerbation of chronic heart failure was the most common cardiovascular disease. The number of nurse home-visits has increased dramatically since FMC opened, and with annual numbers increasing from 64 in FY2018 to 315 in FY2020.

## 4. Discussion

During an emergency phase, an overburdened or destroyed health system is unable to respond appropriately and its ability to “promote, restore or maintain health” is compromised, resulting in an increase in morbidity and mortality [10]. Therefore, immediately after the Fukushima Daiichi NPP accident, there was a focus on the emergency medical needs of workers engaged in the emergency response to the accident. To meet these needs, a temporary emergency medical facility was installed in the Fukushima Daiichi NPP. In addition, a temporary radiation emergency medical facility was established by renovating a medical clinic at the J-Village Soccer Training Facility, located at the boundary of the evacuation order area 20 km south of the NPP [11].

Since 2014, when the evacuation orders for municipalities in Futaba County began to be lifted, the medical needs of returning residents have increased and become more diverse. The ambulance data and our results showed different trends across municipalities, depending on the area, and the time when the evacuation order was lifted. For example, the evacuation orders for residents of Naraha, Tomioka, and Namie Town were lifted by 2017, and the medical needs of the returning residents increased steadily thereafter. Even in Okuma Town, where the evacuation order was lifted the latest, and only for a limited area, the number of patients treated at FMC increased rapidly in FY2020. No major changes were observed in Hirono and Kawauchi; the former was not under an evacuation order, and in the latter only part of the village was evacuated.

The establishment of FMC made it possible to understand the area’s medical needs in greater detail. After FMC was opened, the number of emergency room patients increased with the resident population. In FY2018, most outpatients were in their 60s, and 30% of them had a health insurance card residence outside Futaba County. Most were workers engaged in decontamination and reconstruction projects. These findings coincide with the reasons for attending the hospital; the biggest group of outpatients with injuries and trauma. The increase in older patients (aged over 80 years) since FY2019 is noteworthy. Moreover, a significant increase in the hospitalization of the older patients has been observed since 2018. While the proportion of patients residing in Futaba County has increased, the number of patients from outside the prefecture has decreased, resulting in an increased weight of medical needs for returning older residents. Ten years after the accident, decontamination and reconstruction projects have settled down, and the medical needs of workers are also less.

In FY2020, there was a decrease in outpatients from November to February, along with a downward trend in the proportion presenting with respiratory diseases. This can be related to the responses to the new coronavirus (SARS-CoV-2), namely, the number of people entering the area was reduced, restrictions on going out were in place, hospital visits were avoided when possible, and strict infection control measures were taken, resulting in the reduction of patient volume and lower levels of infectious diseases, including respiratory infections [12].

Although the absolute numbers were small, the number of patients classified into ICD groups E and F increased. The increase in E may have been influenced by an increased prevalence of diabetes due to changes in lifestyle caused by evacuation and long-term dislocation [13]. F also increased, which can reflect the mental health status of returnees [14]. According to Orui et al., many evacuees still have high levels of psychological distress, post-traumatic stress, and health anxiety due to radiation, similar to those who remain evacuated even if their living environment improves [15]. Maeda and Oe also reported serious mental health problems in evacuees, including an increase in disaster-related suicides, despite a gradual decline in anxiety and depressive symptoms [16]. Patients with post-traumatic stress disorder often experience dissociative symptoms and somatization disorders, which may be reflected in the number of F4 cases. However, this study alone is insufficient to make this determination, necessitating the monitoring of this trend in the future.

In Futaba County, family structures changed significantly from predominantly multi-generational families to older couples after the accident. The underdevelopment of public transport makes it difficult for the older people to access medical care, possibly explaining why nurse home-visits increased five-fold. The number of older people with terminal illnesses, e.g., cancer, who wish to be cared for in their hometowns is also increasing, suggesting that the demand for home-visit nursing will continue to increase.

After the acute phase of response to an emergency is over, the health system must recover or rebuild. This represents an opportunity to create a more resilient and fit-for-purpose health system [17] that promotes and safeguards population health and global health security, advances progress towards universal health coverage [18], and plays a central role in the building of community resilience. Creating fit-for-purpose health systems requires an effective, well planned, and well implemented recovery strategy. Post-emergency recovery is, however, usually a missed opportunity to build back better because of lack of knowledge and expertise, inadequate planning, low funding, and competing sociopolitical interests [6].

Futaba County experienced the evacuation of whole towns and villages after a nuclear accident. The return of residents did not proceed as planned because it took time to lift the evacuation order, and there was persistent anxiety about the damaged NPP and radiation [19]. Even now, 10 years after the accident, the population of Futaba County is only about 20% of the pre-accident levels. At least half of the residents of Futaba County have decided not to return [20]. Meanwhile, the government has started a large-scale project to promote the reconstruction of Futaba County, including via the Innovation Coast Initiative. These efforts are expected to bring in a new younger generation, but the future is still uncertain.

Taking the building back better approach to health systems recovery, ensures that the rebuilt system is stronger, safer, smarter, and more resilient [6]. This necessitates the identification and rectification of weaknesses inherent in the previous system. During health systems recovery, health service provision can be improved by addressing previously neglected areas, such as mental health and noncommunicable diseases, strengthening linkages between primary and secondary health services, reviewing the distribution of facilities against demographic changes, and strengthening medical capabilities in areas with the greatest need [21,22]. In Futaba County, patient data have been analyzed monthly by FMC since its establishment, and ambulance transport data are evaluated annually. Health issues of returning residents, including mental health, are monitored by the municipalities and shared among related organizations on a regular basis. The national government conducts a questionnaire survey regularly on the return of residents, and predicts population trends in the region [20]. This information is enabling a study panel to examine the status and issues of the medical systems, including long-term care for older and disabled people, human resources, and financial support. These panels include the national government, Fukushima Prefecture, municipalities, and medical experts in the affected areas. The panels develop specific plans to improve medical systems for Futaba County [7].

The limitations of this study include the following: it focuses on only emergency medical needs, there are no data about patients who visited clinics in Futaba County or other medical institutions outside the county, and ICD classification data were only available for the period after the establishment of FMC. The flow of people in Futaba County is also continuously changing, making it difficult to understand the number of people who require medical care. Although the results of this study reflect only a part of the medical needs of the residents in Futaba County, it is nonetheless valuable information. As the influx of people to this area settles down, a cohort survey may become feasible, and can provide a more accurate understanding of the health status of the residents and related issues after a major nuclear power plant accident.

## 5. Conclusions

Futaba County was the area most affected by the 2011 Fukushima Daiichi NPP accident. As the evacuation orders began to be lifted in 2014, the population of returnees gradually increased. The proportion of ambulance use related to trauma from occupational and traffic accidents increased dramatically, and ambulance use increased at an annual rate of about 10% until 2019. Since 2018, the proportion of ambulance use following traffic accidents has decreased, but the proportion of transport related to endogenous diseases among returned older people has increased. Medical needs in the areas affected by the nuclear accident changed drastically before and after the evacuation order was lifted. The types of illnesses and injuries were related to both the return of the residents, and ongoing decontamination and reconstruction projects. The medical system in the affected area after a nuclear accident must be able to meet these changing and diverse medical demands.

## Figures and Tables

**Figure 1 ijerph-19-00144-f001:**
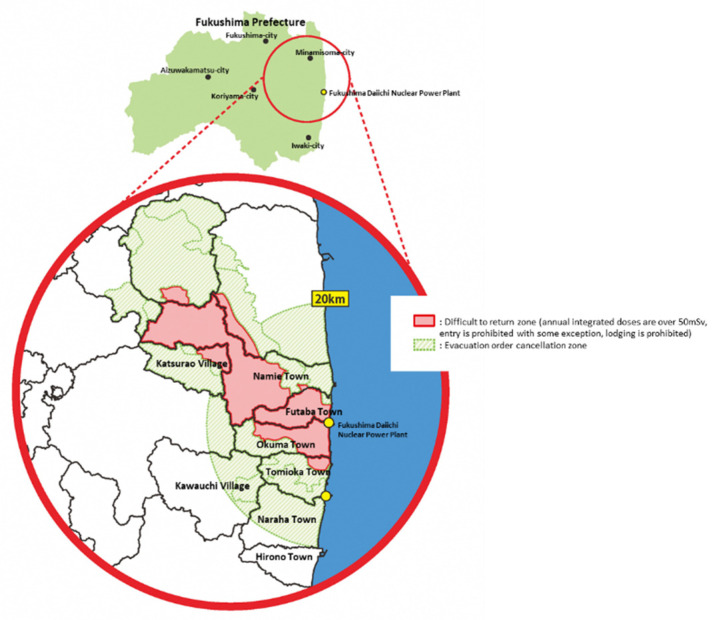
Evacuation order cancellation zone and difficult to return zone, as of April 2021. Red area: difficult to return zone (annual integrated doses are over 50 mSv, entry is prohibited with some exceptions, and lodging is prohibited); and green area: evacuation order cancellation zone.

**Figure 2 ijerph-19-00144-f002:**
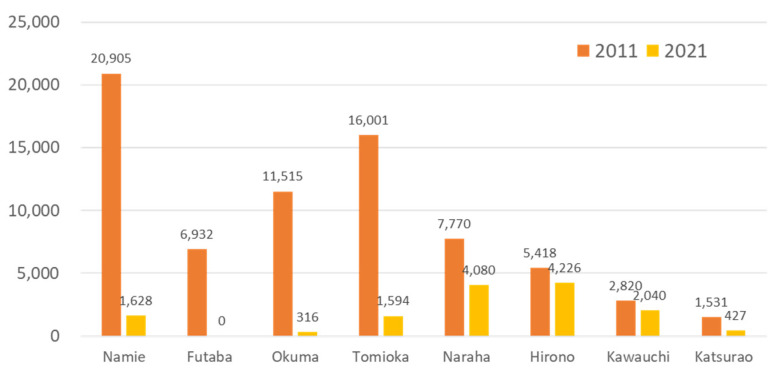
Population of municipalities in Futaba County before (2011) and 10 years after the Fukushima accident (2021).

**Figure 3 ijerph-19-00144-f003:**
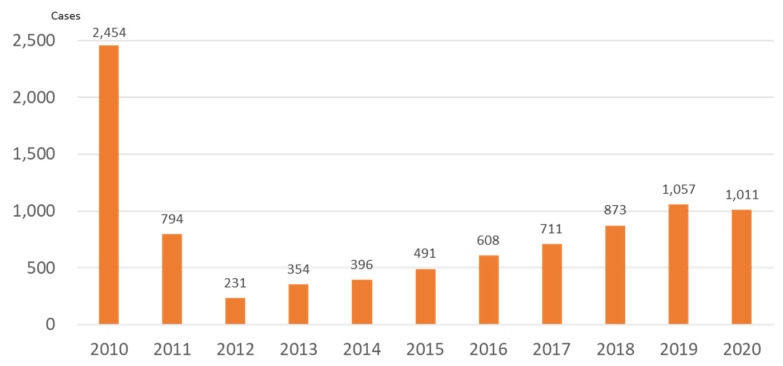
Ambulance transports by the Futaba Fire Department before and after the Fukushima accident (2011).

**Figure 4 ijerph-19-00144-f004:**
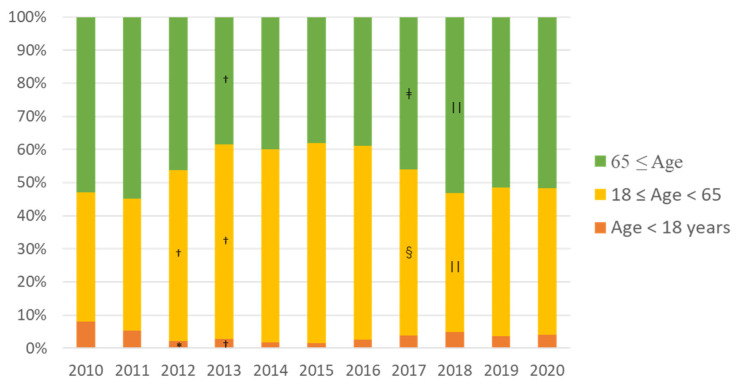
Proportion in each age cohort transported by ambulance before and after the Fukushima accident (2011). * *p* = 0.001, † *p* < 0.001 vs. 2010; ‡ *p* = 0.011, § *p* = 0.003, || *p* < 0.001 vs. 2016. The number of young (<18 years old) and older people (≥65) decreased after the accident compared with before the earthquake (* and †), but, after 2017, when the return of the residents of Namie and Tomioka Town began, the proportion of older people increased compared with 2016 (‡, ||).

**Figure 5 ijerph-19-00144-f005:**
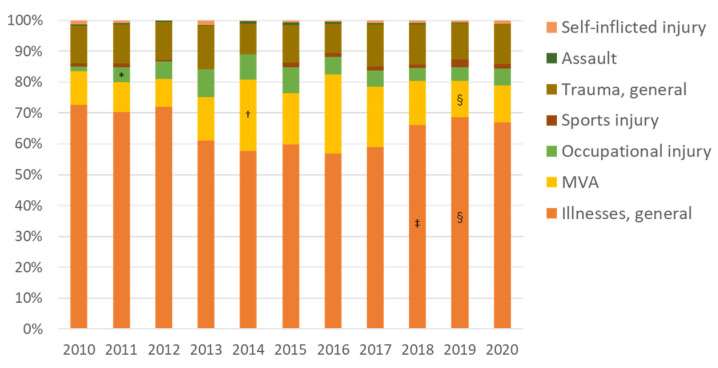
Proportion of injuries and illnesses among patients transported by ambulance before and after the Fukushima accident (2011). MVA, motor vehicle accident. * *p* < 0.001 vs. 2010; † *p* = 0.002 vs. 2013; ‡ *p* = 0.006, § *p* < 0.001 vs. 2017. The proportion of occupational injuries increased four-fold in 2011 compared with before the accident (*). Traffic injuries increased after the evacuation order of Kawauchi Village was lifted in 2014 (†). In 2018, the proportion of endogenous diseases increased compared with 2017 (§).

**Figure 6 ijerph-19-00144-f006:**
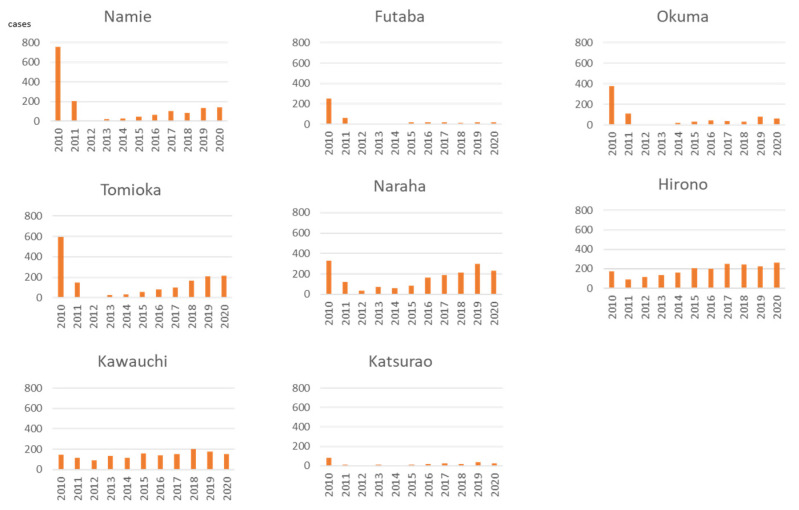
Use of ambulances by municipalities before and after the Fukushima accident (2011).

**Figure 7 ijerph-19-00144-f007:**
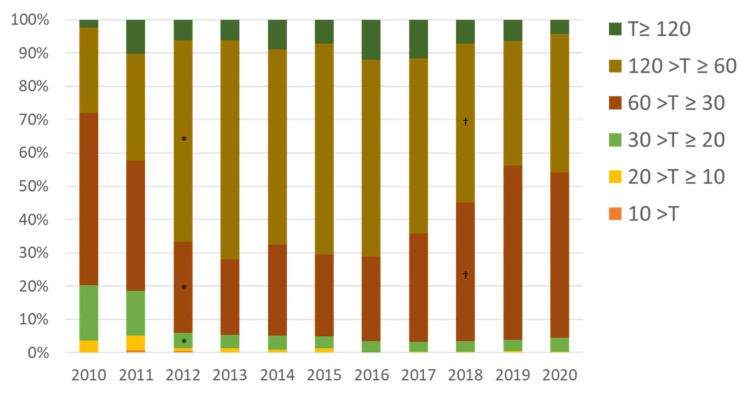
Changes in proportion of time to hospital arrival by ambulance before and after the 2011 Fukushima accident. T is in minutes. * *p* < 0.001 vs. 2010; † *p* < 0.001 vs. 2017. The proportion taking 60 to 120 min increased after the accident (*), but decreased after the Futaba Medical Center was established in 2018 (†).

**Figure 8 ijerph-19-00144-f008:**
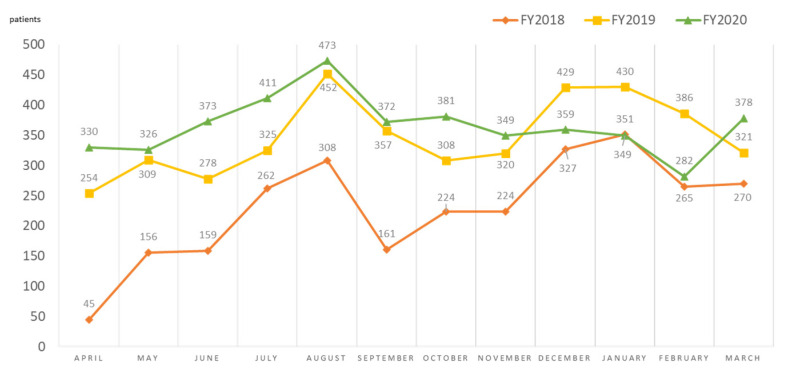
Trends in the monthly number of patients at Futaba Medical Center, FY2018–FY2020.

**Figure 9 ijerph-19-00144-f009:**
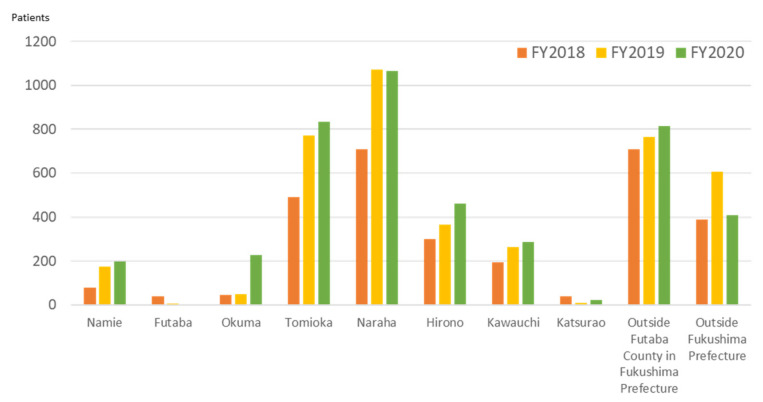
Annual number of patients treated at Futaba Medical Center by municipalities of residence, 2018–2020.

**Figure 10 ijerph-19-00144-f010:**
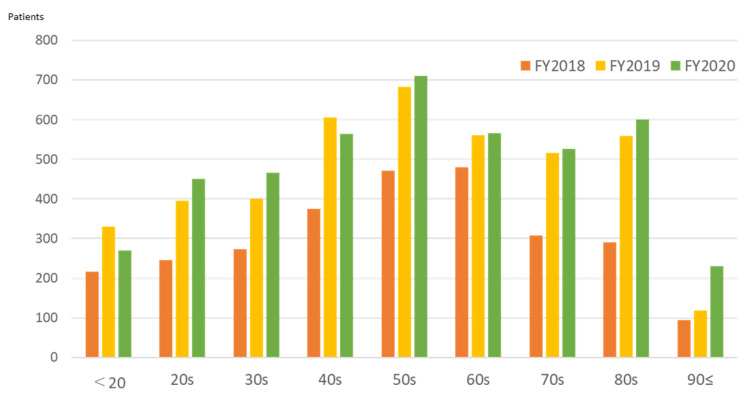
Annual number of outpatients treated at Futaba Medical Center by age cohort (years).

**Figure 11 ijerph-19-00144-f011:**
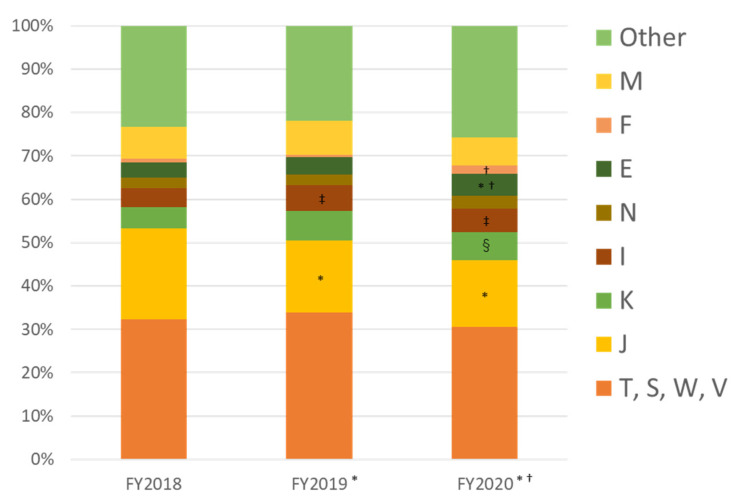
Proportion of outpatients by ICD classification, FY2018–FY2020. * *p* < 0.001 vs. 2018; † *p* < 0.001 vs. 2019; ‡ *p* = 0.005, § *p* = 0.002 vs. 2018. The proportions of J decreased, but E, I, and K increased compared with FY2018 (*, †, §). The proportion of E increased compared with FY2019 (†). ICD: International Classification of Diseases. (M: diseases of the musculoskeletal system and connective tissue; F: mental and behavioral disorders; E: endocrine, nutritional, and metabolic diseases; N: diseases of the genitourinary system; I: diseases of the circulatory system; K: diseases of the digestive system; J: diseases of the respiratory system; S, T: injury, poisoning, and certain other consequences of external causes; and V, W: external causes of morbidity and mortality).

**Figure 12 ijerph-19-00144-f012:**
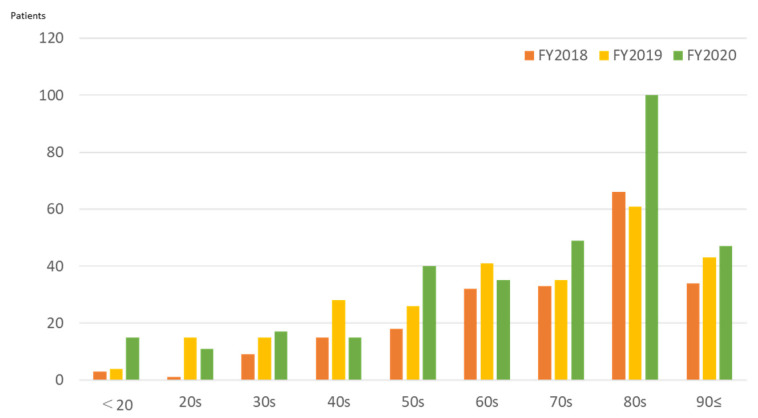
Number of inpatients treated at Futaba Medical Center by age cohort, FY2018–FY2020.

**Figure 13 ijerph-19-00144-f013:**
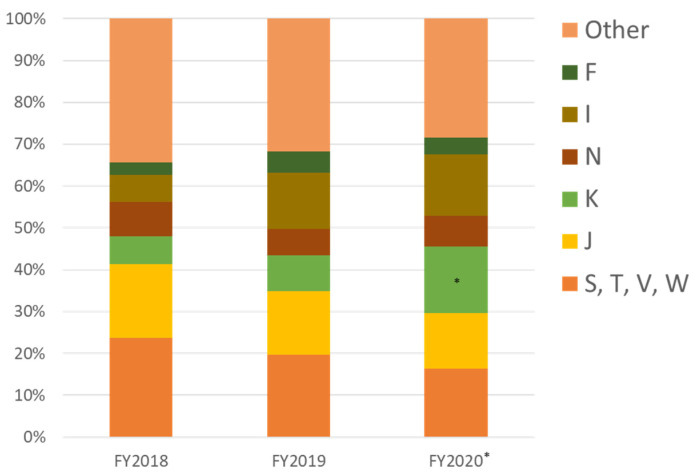
Proportion of inpatients treated at Futaba Medical Center by ICD classification, FY2018–FY2020. * *p* = 0.003 vs. 2018. The proportion of K increased compared with FY 2018. ICD: International Classification of Diseases. (F: mental and behavioral disorders; I: diseases of the circulatory system; N: diseases of the genitourinary system; K: diseases of the digestive system; J: diseases of the respiratory system; S, T: injury, poisoning, and certain other consequences of external causes; and V, W: external causes of morbidity and mortality).

## Data Availability

The data presented in this study are available on request from the corresponding author. The data are not publicly available due to the hospital policy on management of patient data.

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
