# Peer review of "Health and Medical Issues in the Area Affected by Fukushima Daiichi Nuclear Power Plant Accident"

_ijerph, 2021, doi:10.3390/ijerph19010144_

Round 1

Reviewer 1 Report

1. In the abstract and method section, the analysis should be more detailed on how the analysis was done. Did you use descriptive analysis, compare analysis, or others? 2. In the introduction, what is this research's novelty or contribution to scientific knowledge is. This paper sound very locally about Japan and did not relate to an international context; for example may I the author relate to the WHO planning emergency or other framework 3. In the discussion, it is a lack of reference.

Author Response

Thank you very much for the valuable comments. Following is our response to the comments.

  1. In the abstract and method section, the analysis should be more detailed on how the analysis was done. Did you use descriptive analysis, compare analysis, or others?

Author’s response:

We have described the analysis in the Method section and revised the graphs as suggested. Thank you very much for the important comments.

  1. In the introduction, what is this research's novelty or contribution to scientific knowledge is. This paper sound very locally about Japan and did not relate to an international context; for example may I the author relate to the WHO planning emergency or other framework 3.

Author’s response:

We have added citations to the WHO documents suggested (references 6, 10, 17) and revised the text as suggested (lines 51–56, 317–325, 335–352) to clarify the novelty of this work, and its contribution to scientific knowledge.

  1. In the discussion, it is a lack of reference.

We have added new references 21 and 22 in addition to those mentioned above.

Reviewer 2 Report

Intervening, innovative work in the field of disaster medicine. 

Some comments:
- DOI is missing for the bibliography
- conclusions are too general, please be more precise
- lack of correlations in statistics 

Author Response

Thank you very much for the valuable comments. Following is our response to the comments.

Some comments:

- DOI is missing for the bibliography

- conclusions are too general, please be more precise

- lack of correlations in statistics

Author’s response:

-We have added DOIs to the bibliography as suggested.

-We have revised the conclusions as suggested (lines 364–370).

-We have carried out some correlation analysis (lines 85-90), and the results are shown in the graphs (graph 4,5,7,11,13).

Reviewer 3 Report

Thank you for the invitation to review the manuscript.

The manuscript titled “Health and medical issues in the area affected by Fukushima Daiichi Nuclear Power Plant Accident” aims to elucidate issues in the re-development of the medical system. The authors analyzed documents and data from 1). reports of the study panels on medical care and welfare in the evacuation area sponsored by Fukushima Prefecture, 2). information on the returning residents reported by 8 towns and villages in Futaba County, 3). annual report data on ambulance transports provided by the Futaba Fire Department, and 4). data on changes in the resident population.

This is an important report summarizing the changes in medical needs after the Daiichi Nuclear Power Plant Accident.

After reviewing the manuscript, please note the general comments:

  1. what is the significance of the report? the reasons why preparing the report on the changes in medical needs after the accident, and issues in the redevelopment of the medical system should be explicitly mentioned in the introduction.

  1. the researchers should discuss what would be done based on the results, e.g. how to meet the medical needs and changes of population based on benchmarking on the comparative services, population ratio, etc.

specific comments:

  1. the methods section can be elaborated with information about the timeframe at which the data were retrieved, the statistical methods used to analyze the data.
  2. as mentioned, the researchers should recommend actions for the changes in medical needs in the discussion section.

Author Response

Thank you so much for the valuable comments. Following is our response to the comments.

General comments

  1. what is the significance of the report? the reasons why preparing the report on the changes in medical needs after the accident, and issues in the redevelopment of the medical system should be explicitly mentioned in the introduction.

Author’s response:

We have added more information to the Introduction (lines 51–56) to clarify the background of this paper. We have also added more about the significance of this report to the Discussion (lines 317–325, 335–352).

  1. the researchers should discuss what would be done based on the results, e.g. how to meet the medical needs and changes of population based on benchmarking on the comparative services, population ratio, etc.

Author’s response:

We described how we developed the medical system to meet the needs of the region after the accident (lines 342–352).

Specific comments:

  1. the methods section can be elaborated with information about the timeframe at which the data were retrieved, the statistical methods used to analyze the data.

as mentioned, the researchers should recommend actions for the changes in medical needs in the discussion section.

Author’s response:

We have revised the text to add information about the data analysis (Line 76–83). We have also described how data feed into discussions about the changes to the medical system in the Discussion (lines 342–352).